# Increased yield of AP-3 by inactivation of *asm25* in *Actinosynnema pretiosum ssp. auranticum* ATCC 31565

Hong Cheng[1,2☯], Guoqing Xiong[1,2☯], Yi Li[2☯], Jiaqi Zhu[2,3], Xianghua Xiong[2], Qingyang Wang[2], Liancheng Zhang[2], Haolong Dong[2], Chen Zhu[2]*, Gang Liu[2]*, Huipeng Chen[2]

1 Institutes of Physical Science and Information Technology, Anhui University, Hefei, China, 2 Academy of Military Medical Sciences, Beijing, China, 3 School of Life Science and Technology, Dalian University, Dalian, China

☯ These authors contributed equally to this work.
* jueliu@sohu.com (GL); huipengchen@yeah.net (HC)

**Data Availability Statement:** All relevant data are within the paper and its Supporting Information files.

## Abstract

Asamitocins are maytansinoids produced by *Actinosynnema pretiosum ssp. auranticum* ATCC 31565 (*A. pretiosum* ATCC 31565), which have a structure similar to that of maytansine, therefore serving as a precursor of maytansine in the development of antibody-drug conjugates (ADCs). Currently, there are more than 20 known derivatives of ansamitocins, among which ansamitocin P-3 (AP-3) exhibits the highest antitumor activity. Despite its importance, the application of AP-3 is restricted by low yield, likely due to a substrate competition mechanism underlying the synthesis pathways of AP-3 and its byproducts. Given that N-demethylansamitocin P-3, the precursor of AP-3, is regulated by *asm25* and *asm10* to synthesize AGP-3 and AP-3, respectively, *asm25* is predicted to be an inhibitory gene for AP-3 production. In this study, we inactivated *asm25* in *A. pretiosum* ATCC 31565 by CRISPR-Cas9-guided gene editing. *asm25* depletion resulted in a more than 2-fold increase in AP-3 yield. Surprisingly, the addition of isobutanol further improved AP-3 yield in the *asm25* knockout strain by more than 6 times; in contrast, only a 1.53-fold increase was found in the WT strain under the parallel condition. Thus, we uncovered an unknown function of *asm25* in AP-3 yield and identified *asm25* as a promising target to enhance the large-scale industrial production of AP-3.

## Introduction

The development of cancer therapy drugs has undergone several generations [1,2]. Antibody-drug conjugates (ADCs) are one of the fastest growing categories of anticancer drugs, characterized by a monoclonal antibody linked to a molecule with antitumor activity to form conjugates [3,4]. Maytansine, first isolated from African shrubs by Kupchan *et al.* in 1972, is capable of suppressing tubulin polymerization during mitosis, thereby exerting a strong proapoptotic effect on cancer cells [5–7]. Due to its strong cytotoxicity, maytansine has been utilized in the

**Funding:** This work was supported by a fund from the National Natural Science Foundation of China (31900669) to H. D.

**Competing interests:** The authors have declared that no competing interests exist.

development of new generations of ADCs [8,9]. However, the natural content of maytansine in plants is too low, making it difficult to satisfy the fast-growing market demand. Ansamitocins, isolated from *Actinosynnema pretiosum*, have antitumor activity similar to that of maytansine and could serve as a precursor of maytansine due to their structural similarity [10–12]. In contrast to maytansine, which can be extracted only from plants, ansamitocins can be easily obtained by the large-scale fermentation of certain bacterial strains, such as *Actinosynnema pretiosum ssp. auranticum* ATCC 31565 (*A. pretiosum* ATCC 31565). Among more than 20 derivatives [13], AP-3 shows outstanding antitumor activities not only through the inhibition of tubulin polymerization but also by activating dendritic cells to enhance antitumor immunity [14–16]. Despite its apparent advantages, the yield of AP-3 from the original strains is still limited.

Traditional mutagenesis and specific gene modification represent two major ways to obtain AP-3 high-yield strains. By traditional methods, high-yield strains can be obtained by physical irradiation- and compound-induced mutagenesis [17,18]. However, mutations are usually accompanied by unexpected randomness, which makes it difficult to achieve precise control of the direction of evolution. In recent decades, with the explosive development of gene editing technology, targeting a certain gene or a batch of tightly related genes to generate high-yield strains for interesting products has become a possibility, and an increasingly attractive one. Liu *et al.* constructed a multigene coexpression system in *Escherichia coli* BW25113 and obtained strains with a high yield of shikimic acid [19]. Singh *et al.* improved artemisinin yield by ectopic expression of β-glucosidase to increase the glandular hair density in *Artemisia annua* [20]. More recently, alone with a set of genes, *asm1-48*, were found to be closely related to AP-3 biosynthesis, strategies that overexpression or inactivation of these genes in host strains have successfully improved AP-3 yield [17,21–23]. Since N-demethylansamitocin P-3 (PND-3), the precursor of AP-3, is regulated by the genes *asm25* and *asm10* to synthesize ansamitocinosides P-3 (AGP-3) AP-3, respectively [24–26], the competitive consumption of the common substrate in the AGP-3 synthesis pathway may dampen the yield of AP-3. However, the effect of *asm25* on AP-3 yield is still unclear. Hence, in this study, we inactivated *asm25* in *A. pretiosum* ATCC 31565 by CRISPR-Cas9-guided gene knockout and determined AP-3 production by high-performance liquid chromatography (HPLC) (Fig 1). Inactivation of *asm25* indeed increased AP-3 yield in *A. pretiosum* ATCC 31565. Additionally, adding isobutanol, which facilitates the synthesis of the C-3 side chain of AP-3 [27], further improved the yield of AP-3 in the *asm25* knockout strain.

## Materials and methods

### Strains and plasmids

*A. pretiosum* ATCC 31565 was obtained from the ATCC (USA) and cultured on solid or in liquid ISP2 medium containing 0.4% yeast extract, 1% malt extract and 0.4% glucose (pH 7.2). Solid medium contains 1.8% agar. DH5α *E. coli* competent cells were purchased from CoWin Biosciences, and *E. coli* electrocompetent cells (ET12567, pUZ8002) were in laboratory stock. The pCas9 (pWHU2653) plasmid used for *asm25* targeting was in laboratory stock as well.

### Construction of the knockout plasmid pCas9-asm25-LR

The sgRNA sequence "ctcgttgagccggggcagca" targeting *asm25* was designed according to AF453501, deposited in NCBI GenBank, and synthesized artificially. A 0.3 kb fragment located between the *PermE* promoter and Terminator was cleaved from pCas9 pWHU2653 with *Nhe*I and *Xba*I, and then the sgRNA was inserted to place the sgRNA under the control of the *PermE* promoter. The fragment containing sgRNA was cloned into pWHU2653, resulting in

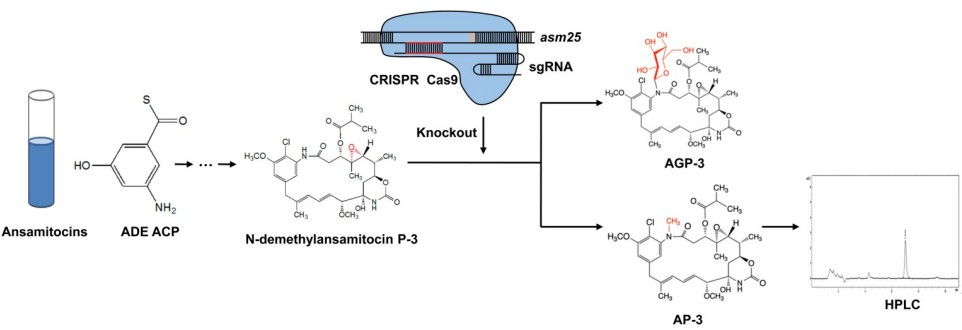

**Fig 1. Schematic of the biosynthesis pathway of AP-3 and a brief workflow.**

K-*asm25*sg. Subsequently, homologous arms flanking *asm25* were amplified from *A. pretiosum* ATCC 31565 genomic DNA and named *asm25*-L (1.4 kb) and *asm25*-R (1.5 kb). Then, *asm25*-LR (2.9 kb) was generated from *asm25*-L and *asm25*-R by overlapping PCR, followed by cloning into K-*asm25*sg to obtain the pCas9-*asm25*-LR targeting plasmid. The primers used for PCR amplification are listed in Table 1. A diagrammatic sketch of the construction of the *asm25* targeting plasmid is shown in S1 Fig.

## Generation of the *asm25* knockout strain

pCas9-*asm25*-LR was introduced into *A. pretiosum* ATCC 31565 via *E. coli-Actinosynnema* biparental conjugation as described [28]. Briefly, the pCas9-*asm25*-LR plasmid was electro-transferred into *E. coli* ET12567 competent cells and then cells were grown on LB solid medium containing kanamycin (25 μg/mL), chloramphenicol (25 μg/mL) and apramycin (100 μg/mL) at 37°C overnight to obtain a single clone. After heat shock, *A. pretiosum* ATCC 31565 cells were resuspended in LB medium and mixed with *E. coli* ET12567, followed by cultivation for 3–7 days on a solid medium with apramycin to obtain exconjugants. To screen for double-crossover mutants, single clones of exconjugant were cultured in liquid medium without the presence of apramycin for three rounds. Then, the exconjugants were plated on solid medium with or without apramycin. The clones that regained sensitivity to apramycin were selected for analysis for *asm25* deletion via PCR amplification with the primers asm25-seq-F and asm25-seq-R (Table 1). Positive clones were further confirmed by sequencing.

## Fermentation of *A. pretiosum* ATCC 31565 in flasks

*A. pretiosum* ATCC 31565 single clones from culture dishes were cultured in seed culture medium (yeast extract: 4 g/L, maltose: 10 g/L, dextrose: 4 g/L and sodium chloride: 1.5 g/L) for 60 hours. Subsequently, 10 mL of seed medium was added to 200 mL of fermentation medium (yeast extract: 4 g/L, maltose: 10 g/L, soluble starch: 4 g/L and sodium chloride: 1.5 g/L) in a

**Table 1. Primers used in this study.**

| Primer | Primer sequence (5'-3') |
| --- | --- |
| asm25-L-F | GCAAGCTTACGTTGTCCACCCCGTTGTCC |
| asm25-L-R | GGTTCTGGGTGGAGCTGGCCGATCCCCGCTTCACC |
| asm25-R-F | GGATCGGCCAGCTCCACCCAGAACCTCGC |
| asm25-R-R | GCAAGCTTTGGCGGACGTGGACGACGACC |
| asm25-seq-F | ACCAGCGGAGGAGGAGACCCA |
| asm25-seq-R | CCCCAGCACGGAGGAAGACAG |

500 mL cone flask at a ratio of approximately 1:20. Fermentation lasted for 9 days, and the medium was sampled every 24 hours for the determination of AP-3 yield. For isobutanol-related experiments, various amounts of isobutanol were added to the broth at the initiation of the fermentation reaction.

### Determination of AP-3 yield by HPLC

After centrifugation, 2 mL of fermentation supernatant was collected and mixed vigorously with the same volume of ethyl acetate. The ethyl acetate phase was collected and mixed with ethyl acetate again for further extraction. After rotary evaporation, the concentrates were redissolved with methanol and subjected to component analysis by HPLC.

An Agilent 1260 Infinity II instrument was used for HPLC analysis. The stationary phase was an Agilent EC-C18 column (4 μm, 4.6 × 250 mm). The mobile phase was 52% acetonitrile/48% water after degassing with a filter membrane and ultrasonication. The configuration was set as follows: the flow rate was 1 mL/min, the wavelength of the UV detector was 254 nm, the column temperature was 25°C, and the injection volume of the sample was 5 μL. The sample was injected automatically. The concentration of AP-3 standard ranging from high to low (80, 40, 20, 10, 5, 2.5, 1.25 and 0.625 mg/L) was prepared by gradient dilution in chromatographically pure methanol. The concentration of AP-3 in the samples was calculated by comparison with the AP-3 standard curve.

### Statistical analysis

GraphPad Prism 9.0.0 (developed by GraphPad Software) was utilized to perform statistical analysis for AP-3 yield comparisons. The data are presented as the mean ± standard deviation (SD). The comparison between *asm25*-knockout and WT used Student's *t* test, whereas the means of the six groups from the isobutanol experiments were compared by ANOVA.

## Results

### Establishment of a method for the detection of AP-3 by HPLC

First, we established an applicable approach to detect AP-3 from fermentation samples, as described above. Based on the principle that the x-axis indicates the concentration of the AP-3 standard and the y-axis indicates an absorbing peak area, the standard curve was generated (Fig 2A). The standard curve equation was $Y = 10.261X - 2.6282$, $R^2 = 0.9999$, and the linear relationship held true within the AP-3 concentration range of 0.625–80 mg/L (Fig 2A). The fermentation sample peaked at the same time point as the AP-3 standard (red arrow, Fig 2B and 2C), suggesting a presence of AP-3 in the sample. Then, the effluent indicated by the peak (red arrow in Fig 2B and 2C) was harvested for further analysis by nuclear magnetic resonance (NMR) and liquid chromatography-mass spectrometry (LS-MS), carried out by Dalian University and the Institute of Process Engineering of Chinese Academy of Sciences, respectively. The product was identified as AP-3 (S2 Fig). Thus, we successfully set up a practicable method to detect AP-3 from fermentation samples.

### Inactivation of *asm25* in *A. pretiosum* ATCC 31565

The diagram describing the construction of the pCas9-a*sm25*-sgRNA targeting plasmid is shown in Fig 3A. Cloning of *asm25*-sgRNA into pWHU2653 did not cause plasmid degradation (Fig 3B). Then, the homologous arms flanking *asm25* (1.5 kb for arm R and 1.4 kb for arm L) were amplified by PCR and then merged into one fragment (2.9 kb) (Fig 3C). After double-crossover screening (S3 Fig), selected exconjugants were analyzed for *asm25*

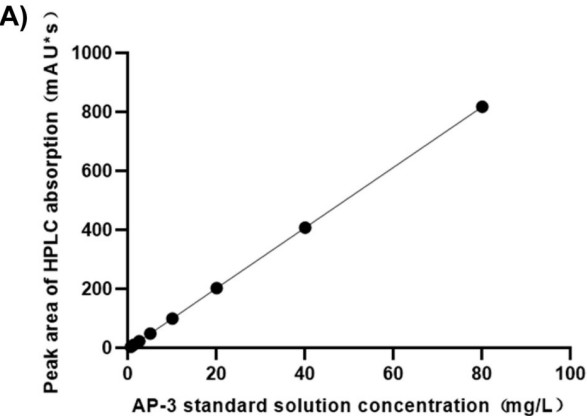

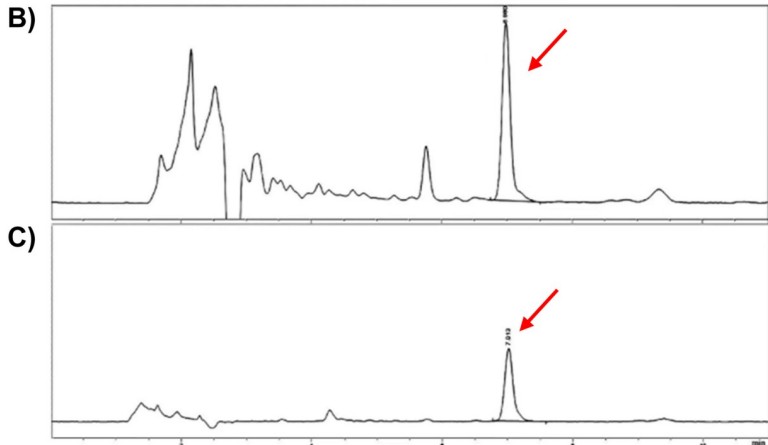

**Fig 2. HPLC analysis of AP-3 production.** (A) AP-3 standard curve. (B) Products of the fermentation sample. (C) AP-3 standard. Red arrows indicate the AP-3 position in the graphs.

depletion. Theoretically, the *asm25* knockout strain gave 0.7 kb products, whereas the wild type gave 1.6 kb products. Thus, clones 2, 5, 6, 7, and 9 were considered to exhibit *asm25* knockout (Fig 3D). Cell sequencing showed that the most part of *asm25* (885 bp of 1209 bp, 132 to 1016) was depleted from *A. pretiosum* ATCC 31565 genome, as expected (S4 Fig). Thus, the *asm25*-inactivated *A. pretiosum* ATCC 31565 strain was successfully established.

## Increased yield of AP-3 in the *asm25* knockout strain

To investigate whether *asm25* knockout might affect AP-3 yield, WT and *asm25*-knockout strains of *A. pretiosum* ATCC 31565 were cultured in parallel fermentation conditions for 9 days. Samples were collected every 24 hours from Day 3 to the end of the fermentation. The levels of AP-3 were determined after all samples were collected. It was shown that *asm25* knockout resulted in more AP-3 production than the WT strain from Day 6, and the difference in AP-3 levels between the two groups increased as fermentation progressed. At the end of the fermentation, the AP-3 yield reached a peak in each group, and the AP-3 yield in the *asm25* knockout strain (4.42 ± 0.58 mg/L) was increased more than 2-fold in comparison with that of the WT strain (1.93 ± 0.15 mg/L) (Fig 4).

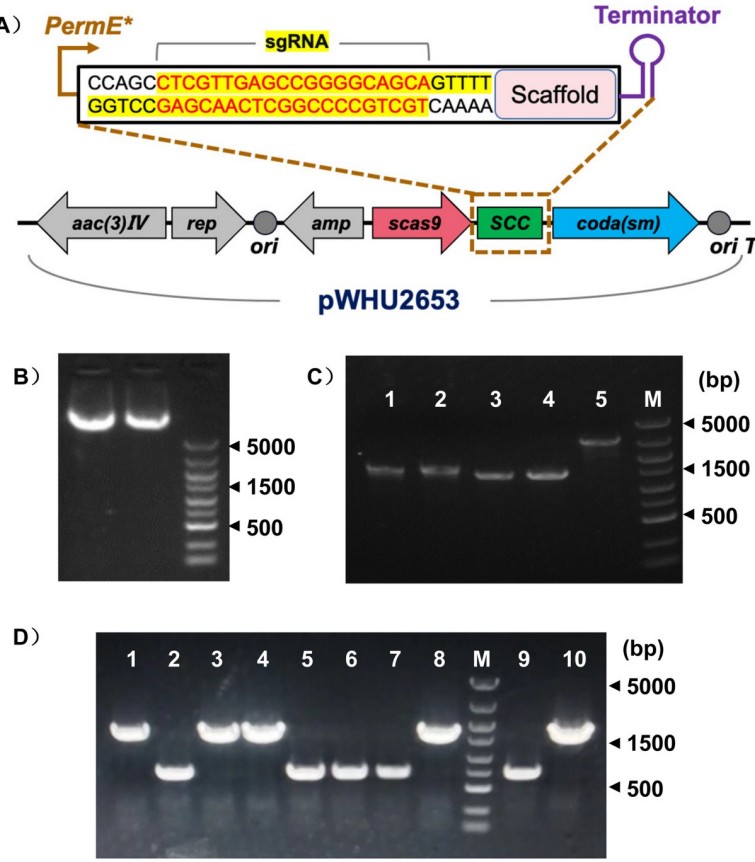

**Fig 3. Generation of the *asm25*-inactivated strain.** (A) Schematic diagram of plasmid construction. Gel electrophoresis of pWHU2653 harboring *asm25* sgRNA (both lanes). Plasmids were purified from *E. coli*. (C) Gel electrophoresis of homologous arms of *asm25*. Lanes 1 and 2 indicate arm R, Lanes 3 and 4 indicate arm L, and Lane 5 indicates arm LR fusion. (D) Gel electrophoresis of selected double-crossover exconjugantes. The primers asm25-seq-F and asm25-seq-R were used for PCR.

## *asm25* inactivation augments the improvement of AP-3 production by isobutanol

The addition of isobutanol has been reported to enhance the yield of AP-3 in fermentation via a C-3 side chain synthesis mechanism [27]. To investigate whether the addition of isobutanol combined with the knockout of *asm25* has a synergistic effect on AP-3 yield, the fermentation processes started with the addition of a variety of concentrations of isobutanol (0, 20, 30, 40, 50 and 60 mM). As expected, adding isobutanol indeed improved AP-3 yield, with isobutanol used at concentrations of 20, 30, 40 and 50 mM. Interestingly, adding 40 mM isobutanol resulted in the highest production of AP-3 (2.92 ± 0.09 mg/L versus 1.39 ± 0.09 mg/L without isobutanol), suggesting an optimal concentration of isobutanol for AP-3 production (Fig 5A). Surprisingly, adding isobutanol showed a marked improvement in AP-3 yield in the *asm25* knockout strain. As shown in Fig 5B, AP-3 yield in the group with 40 mM isobutanol was dramatically improved during the fermentation process, up to 27.43 ± 1.03 mg/L at Day 9, increasing almost 6-fold in comparison with that without isobutanol (4.42 ± 0.23 mg/L) and 9-fold compared with that in the WT strain with isobutanol (2.92 ± 0.09) (Fig 5A and 5B).

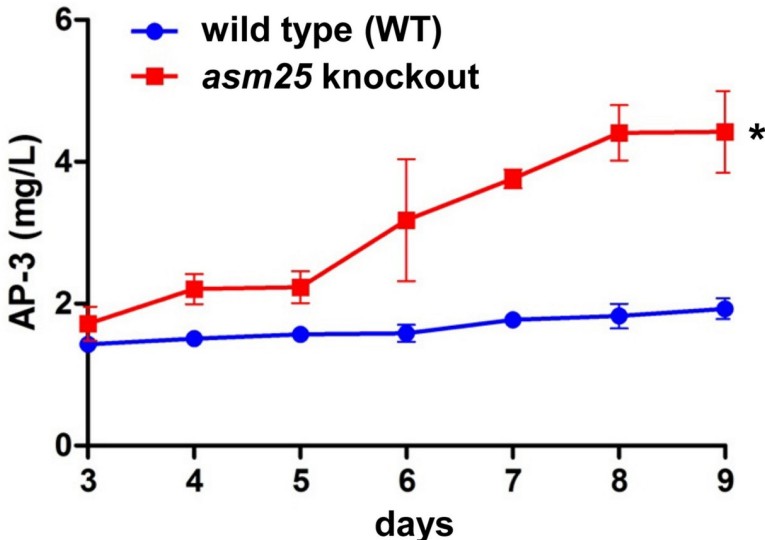

**Fig 4. Determination of AP-3 yield in *asm25* knockout and WT strains by HPLC.** Student's *t* test was used for statistical analysis. *: *p*<0.05. The mean ± SD from three independent experiments (n = 3) are presented. Blue line: WT strain; Red line: *asm25* knockout strain.

## Discussion

Ansamitocins, known as the microbial versions of natural maytansinoids, represent a family of 19-membered macrocyclic lactams with outstanding cytotoxic and antitumor activities, which are widely used to design ADCs [29]. According to different side-chain substituents, ansamitocins can be roughly classified into 20 types, among which AP-3, with an isobutyryl group connected to the side chain at the C-3 position of the mother nucleus, exhibits a similar chemical structure and antitumor activity to maytansine. Currently, AP-3 is mainly obtained from biological fermentation. However, the yield of AP-3 is still hampered by the low productivity of its host strains, i.e., *A. pretiosum* ATCC 31565. In recent years, the genes belonging to the ansamitocin biosynthesis gene cluster (*asms*) have been functionally determined, and with the rapid development of synthetic biology technology, the situation has substantially improved. Pan *et al*. identified *asm8* as an AP-3-positive gene and demonstrated that ectopic overexpression of *asm8* improved AP-3 yield [21]. Ning *et al*. inactivated *asm30* (negative AP-3) and overexpressed *asm10* (positive AP-3) in *A. pretiosum subsp. pretiosum* ATCC 31280, obtaining a strain with a high AP-3 yield [22]. Hence, manipulation of the genes tightly related to AP-3 biosynthesis seems to be a promising way to obtain AP-3 high-yield strains.

In our study, *asm25* was chosen for investigation of its relationship with AP-3 yield. *asm*25 is known as an N-glycosyltransferase of ansamitocin that catalyzes the N-glycosylation of PND-3 to generate AGP-3 [26]. Because PND-3 is the common precursor for both AGP-3 and AP-3, we hypothesized that *asm25* inactivation might shift the biological reaction to AP-3 synthesis. As expected, the AP-3 yield doubled upon *asm25* knockout without affecting the general biological features of the host strain. To our surprise, in the presence of isobutanol, a supplement known to be advantageous to AP-3 biological synthesis, the yield of AP-3 increased more than 9-fold in the *asm25* knockout strain compared with the WT. These data suggest that not only *ams25* inactivation has a synergistic effect with isobutanol on AP-3 yield, but inactivation of *asm25* also enlarges the effect, resulting in a substantial increase in AP-3 yield. Notably, *asm25* knockout also improved the production of some byproducts, AP-2 and AP-4, by more than 10-fold (6.33 ± 1.23 mg/L versus 0.59 ± 0.19 mg/L in WT) and 1.2-fold

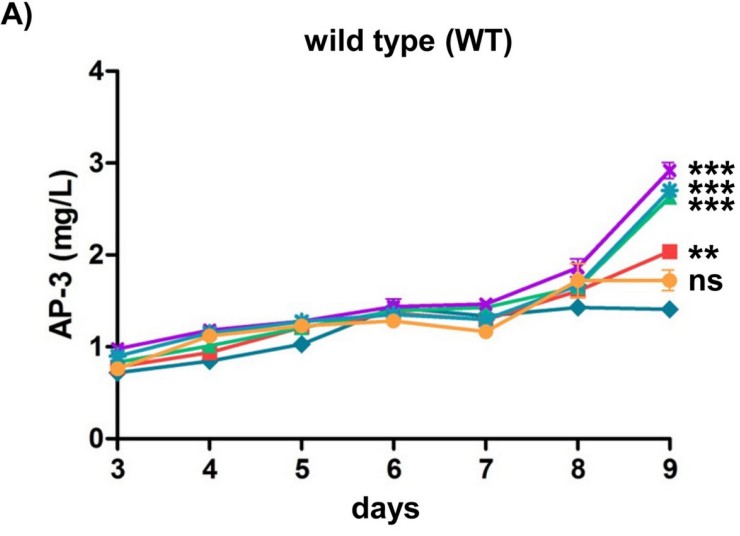

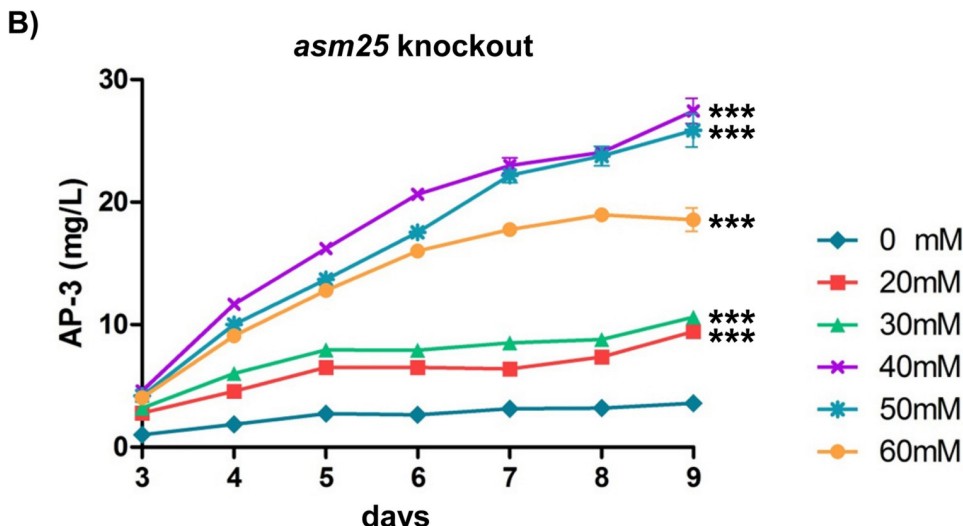

**Fig 5. Determination of AP-3 yield in the presence of isobutanol.** (A) AP-3 yield curve in the WT strain. (B) AP-3 yield curve in the *asm25* knockout strain. Lines in color represent various concentrations of isobutanol in fermentation. The mean ± SD from three independent experiments (n = 3) are presented. The comparison and statistical analysis for AP-3 yield at Day 9 among groups were performed by the ANOVA. ∗∗: $p < 0.01$; ∗∗∗: $p < 0.001$.

(2.09 ± 0.17 mg/L versus 1.88 ± 0.23 mg/L in WT), respectively (S5 Fig). This is not puzzling because in addition to AGP-3, *asm25* is also responsible for the catalysis of PND-2 and PND-4 to generate AGP-2 and AGP-4, respectively [26], logically resulting in the inhibition of AP-2 and AP-4 synthesis. Unfortunately, because the AGP-3 standard was not available, we did not perform assays to quantify the AGP-3 yield in the *asm25* knockout strain. Further experiments should be carried out to address this issue to better understand the mechanisms by which *asm25* regulates AP-3 yield.

Several attempts have successfully improved the biological yield of AP-3 via a variety of strategies. Traditionally, a high yield of AP-3 can be obtained from physical and chemical factor-induced mutagenesis in host strains. Previously, a random mutagenesis approach was established by Chung *et al.* to generate mutant strains that produce 5- to 10-fold more AP-3

than the parental strain [30]. Du *et al.* isolated a mutant strain with a 3-fold AP-3 yield increase by an N-methyl-N'-nitro-N-nitrosoguanidine mutation [17]. Through mutation breeding, Zhong *et al.* generated a high AP-3 yield *A. pretiosum* ATCC 31565, in which the production of AP-3 increased up to 19.7-fold [18]. Although mutagenesis has shown advantages in obtaining new strains, completely random mutations in the host genome critically dampen the subsequent screening efficiency. Hence, targeting certain genes responsible for AP-3 synthesis has become a more popular way to generate strains with high AP-3 yield. The results from Ning *et al.*, as mentioned above, showed that inactivation of *asm30* and overexpression of *asm10* improved AP-3 yield by 66% and 93%, respectively. In another study, overexpression of *asm13-17* improved the AP-3 titer in a shake flask by 1.94-fold. In our study, by contrast, inactivation of *asm25* exhibited a more profound effect on AP-3 production, with a more than 9-fold increase compared with WT, in the presence of 40 mM isobutanol. Although the absolute titer of AP-3 in our study was not remarkably high, optimization of the fermentation conditions and application of a fed-batch culture strategy might give rise to a more profound improvement in AP-3 yield in the *asm25*-inactivated strain. Collectively, we demonstrated that *asm25* is a novel regulator of AP-3 yield. It is of great interest that involving *asm25* inactivation in existing high-yield strains might further improve AP-3 yield to a new level.

## Supporting information

**S1 Fig. Work flow of the construction of pCas9-*asm25*-LR, the *asm25* targeting plasmid.** (PDF)

**S2 Fig. Determination of the fermentation products in *A. pretiosum ssp. auranticum* ATCC 31565.** The products were analyzed and identified by Waters 600 series high performance liquid chromatograph (HPLC) and Bruker microtof-q II mass spectrometer (MS). The HPLC chromatogram and MS spectrum were well consistent with each other. Three samples, sample A, B and C, whose peak time ranged from 15 min to 30 min in MS spectrum, were further analyzed by hydrogenation analysis and compared with related database. Sample A, B and C were identified as AP-2, AP-3 and AP-4, respectively. (A) HPLC chromatograms. (B) MS spectrums. (PDF)

**S3 Fig. Screening for *asm25* knockout strains.** (A) Upper row: *A. pretiosum ssp. auranticum* ATCC 31565 cells mixed with plasmid-transformed *E. coli* ET12567 competent cells were plated on solid medium containing apramycin. Exconjugates were visible after 3–7 days. Lower row: Exconjugates single clones were sub-cultured for continuous three times on ISP2 solid medium without antibiotics to obtain knockout strain. (B) Exconjugants were plated on solid medium with or without apramycin to make the double-crossover clones visualized. Possible *asm25* knockout strains were indicated by the red circles, which were further confirmed by PCR analysis. (PDF)

**S4 Fig. Sequencing of *asm25* gene in the *A. pretiosum ssp. auranticum* ATCC 31565 *asm25* knockout strain.** (A) Alignment between sequencing result (from asm25-seq-R) and AF453501 (*Actinosynnema pretiosum subsp. auranticum* maytansinoid antitumor agent ansamitocin biosynthetic gene cluster I, partial sequence, 82746 bp). A fragment of 885 bp was depleted from *asm25* gene (885 bp of 1209 bp). (B, C) Alignment in detail by zooming in A. (PDF)

**S5 Fig. HPLC spectrum of the fermentation products of ansamitocins.** Red arrows showed
AP-3, Blue arrows showed AP-2 and AP-4.
(PDF)

## Author Contributions

**Conceptualization:** Huipeng Chen.

**Data curation:** Xianghua Xiong.

**Funding acquisition:** Haolong Dong.

**Investigation:** Hong Cheng, Guoqing Xiong.

**Methodology:** Jiaqi Zhu.

**Project administration:** Liancheng Zhang, Chen Zhu.

**Resources:** Liancheng Zhang, Chen Zhu.

**Supervision:** Gang Liu.

**Validation:** Yi Li.

**Writing – original draft:** Hong Cheng, Jiaqi Zhu.

**Writing – review & editing:** Qingyang Wang.

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
