## [Decision Letter · Decision Letter 0]

12 Jan 2022

PONE-D-21-32320Increased Yield of AP-3 by Knockout of asm25 in Actinosynnema pretiosum ssp.auranticum ATCC31565PLOS ONE

Dear Dr. Chen,

Thank you for submitting your manuscript to PLOS ONE. After careful consideration, we feel that it has merit but does not fully meet PLOS ONE’s publication criteria as it currently stands. Therefore, we invite you to submit a revised version of the manuscript that addresses the points raised during the review process.

Your manuscript was reviewed by an expert. They think that it will not be accepted without a major revision(s). Please revise it according to their suggestions. In addition, please also find my suggestions in the “Comments to Author” section.

We look forward to receiving your revised manuscript.

Kind regards,

Hodaka Fujii, M.D., Ph.D.

Academic Editor

PLOS ONE

Journal Requirements:

Additional Editor Comments (if provided):

1. Professional English language editing must be done as the reviewer suggested.

2. Line 161: Section 2.3 is missing.

3. Line 164 & 173: “Data not shown” is not acceptable. Please provide data in the main body or as Supplementary data.

4. Fig. 6: Student’s t-test cannot be used for comparison of three or more data sets. ANOVA must be used. Labeling of (A) WT and (B) mutant in the figure will make it more understandable.

Reviewers' comments:

Reviewer's Responses to Questions

**Comments to the Author**

1. Is the manuscript technically sound, and do the data support the conclusions?

Reviewer #1: Partly

2. Has the statistical analysis been performed appropriately and rigorously? 

Reviewer #1: Yes

3. Have the authors made all data underlying the findings in their manuscript fully available?

Reviewer #1: No

4. Is the manuscript presented in an intelligible fashion and written in standard English?

Reviewer #1: No

5. Review Comments to the Author

Reviewer #1: 1. In-depth experimental design and result analysis should be performed. It’s suggested the quantitative research of substrate consumption and by-products accumulation in the competent pathway(s) be provided, and the productivity be compared with previous research.

2. The gene sequencing results should be given to show the details of indels to prove the effectiveness of the CRISPR/Cas system.

3. The paragraph describing the plasmid/strain construction is best to precede other parts in the section of results.

4. The section of Materials and Methods should be given concisely and accurately, and some obvious errors must be corrected. For example, the concentration of antibiotics used for procaryotic cells usually ranks around mg/L, not g/L.

5. The writing is not acceptable for publication at present for a bundle of problems with sentence structure, verb tense, and clause construction (…Hence, 243 manipulation the genes related to AP-3 biosynthesis by the novel synthetic biology technology is 244 proven a feasible way to obtained high AP-3-yield strains…). It’s strongly suggested the authors have the work reviewed by a proper translation/reviewing service before submission.

6. PLOS authors have the option to publish the peer review history of their article (what does this mean?). If published, this will include your full peer review and any attached files.

Reviewer #1: No

---

## [Author Response · Author response to Decision Letter 0]

11 Feb 2022

Dear editor,

We thank you for giving us opportunity to revise this paper. We carefully read the view letter and have made our best effort to perform paper revision. We also thank you and reviewer for finishing reading our manuscript with good patience. We are answering the questions as followed:

Reviewer #1:

1. In-depth experimental design and result analysis should be performed. It’s suggested the quantitative research of substrate consumption and by-products accumulation in the competent pathway(s) be provided, and the productivity be compared with previous research.

Answer: We thank reviewer for this suggestion. To answer it, we determined the yield of AP-2 and AP-4, two byproducts, in the asm25 knockout strain, and compared their yield with that of the WT strain. Results were described in the discussion section. Next, we have gotten some AGP-3 standards from other labs and tried quantification experiments. Unfortunately, these standards were not qualified for perform HPLC analysis, and there is no commercial AGP-3 available. Thus, the determination of AGP-3 accumulation might be carried out in the further investigation. Last, we compared the effects of asm25 inactivation in our study and other strategies from previous research, on the productivity of AP-3, and discussed this issue in the discussion section. 

2. The gene sequencing results should be given to show the details of indels to prove the effectiveness of the CRISPR/Cas system.

Answer: We thank reviewer for this suggestion. Gene sequencing results of asm25 knockout strain has been shown in Fig. S4. 

3. The paragraph describing the plasmid/strain construction is best to precede other parts in the section of results. 

Answer: We thank reviewer for the suggestion. We rewrote some parts in the Materials and Methods and Results sections, to make the description of the construction more clear. In this version, the construction of the plasmid was described in the Materials and Methods section and summarized in Fig. S1, while the generation of asm25 knockout strain was described in the Results section. Screening for the exconjugants was shown in Fig. S3. Sequencing for the asm25 knockout strain was shown in Fig. S4. Consistently, the panels involved in previous Fig. 3 and Fig. 4 were combined and constituted new Fig. 3.

4. The section of Materials and Methods should be given concisely and accurately, and some obvious errors must be corrected. For example, the concentration of antibiotics used for procaryotic cells usually ranks around mg/L, not g/L.

Answer: We sincerely thank the reviewer for this suggestion. We have rewritten the whole section of Materials and Methods to make it easier to understand, and corrected all obvious errors.

5. The writing is not acceptable for publication at present for a bundle of problems with sentence structure, verb tense, and clause construction (…Hence, 243 manipulation the genes related to AP-3 biosynthesis by the novel synthetic biology technology is 244 proven a feasible way to obtained high AP-3-yield strains…). It’s strongly suggested the authors have the work reviewed by a proper translation/reviewing service before submission

Answer: We feel sorry for the unprofessional English writing. A language editing has been done by the Springer Nature Service to make it read more fluently. 

Additional comments from editor:

1. Professional English language editing must be done as the reviewer suggested.

 Answer: We feel sorry for that. A language editing has been done by the Springer Nature Service to make it read more fluently.

2. Line 161: Section 2.3 is missing.

 Answer: We have made the correction by renumber the section.

3. Line 164 & 173: “Data not shown” is not acceptable. Please provide data in the main body or as Supplementary data.

Answer: We thank editor for this suggestion. The identification of AP-3 (Line 164) was described in the first graph in the Results section, and the results of MS analysis were shown in Fig. S2. The sequencing results of the asm25 knockout strain were shown in Fig. S4.

4. Fig. 6: Student’s t-test cannot be used for comparison of three or more data sets. ANOVA must be used. Labeling of (A) WT and (B) mutant in the figure will make it more understandable.

 We appreciated that editor gave such a good suggestion. ANOVA has been used to reanalyze the data from six different groups in isobutanol experiments. Annotations have been added to Fig. 5 to indicate the WT and asm25 knockout strain, respectively.

---

## [Editor Report · Decision Letter 1]

3 Mar 2022

Increased Yield of AP-3 by Inactivation of asm25 in Actinosynnema pretiosum ssp. auranticum ATCC31565

PONE-D-21-32320R1

Dear Dr. Chen,

We’re pleased to inform you that your manuscript has been judged scientifically suitable for publication and will be formally accepted for publication once it meets all outstanding technical requirements.

Kind regards,

Hodaka Fujii, M.D., Ph.D.

Academic Editor

PLOS ONE
---

## [Editor Report · Acceptance letter]

10 Mar 2022

PONE-D-21-32320R1 

Increased Yield of AP-3 by Inactivation of *asm25* in *Actinosynnema pretiosum ssp. auranticum* ATCC31565 

Dear Dr. Chen:

I'm pleased to inform you that your manuscript has been deemed suitable for publication in PLOS ONE. Congratulations! Your manuscript is now with our production department. 

Kind regards, 

on behalf of

Dr. Hodaka Fujii 

Academic Editor

PLOS ONE